# Care Bundles after Discharging Patients with Chronic Obstructive Pulmonary Disease Exacerbation from the Emergency Department

**DOI:** 10.3390/medsci6030063

**Published:** 2018-08-07

**Authors:** Elisenda Gómez-Angelats, Carolina Sánchez

**Affiliations:** 1Emergency Department, Hospital Clínic de Barcelona, C/Villarroel 170, 08036 Barcelona, Spain; 2Emergencies Group: Processes and Pathologies, Institut d’Investigacions Biomèdiques August Pi i Sunyer, C/Rosselló 149, 08036 Barcelona, Spain; csanchezm@clinic.cat

**Keywords:** acute exacerbation of chronic obstructive pulmonary disease, chronic obstructive pulmonary disease, COPD, care bundle, emergency health services

## Abstract

Chronic obstructive pulmonary disease (COPD) is the second leading cause of emergency department (ED) admissions to hospital, and nearly a third of patients with acute exacerbation (AE) of COPD are re-admitted to hospital within 28 days after discharge. It has been suggested that nearly a third of COPD admissions could be avoided through the implementation of evidence-based care interventions. A COPD discharge bundle is a set of evidence-based practices, aimed at improving patient outcomes after discharge from AE COPD; body of evidence supports the usefulness of discharge care bundles after AE of COPD, although there is a lack of consensus of what interventions should be implemented. On the other hand, the implementation of those interventions also involves different challenges. Important care gaps remain regarding discharge care bundles for patients with acute exacerbation of COPD discharged from EDs There is an urgent need for investigations to guide future implementation of care bundles for those patients discharged from EDs.

## 1. Introduction

Acute exacerbations of chronic obstructive pulmonary disease (AECOPD) account for an appreciable number of acute medical admissions, and a significant proportion of these patients are admitted via emergency departments (EDs). As well as being an important cause of unplanned admissions, COPD is the second leading cause of ED admissions to hospital [1].

Exacerbations become more frequent and severe as the disease progresses; moreover, the progressive nature of COPD and predisposition to exacerbations results in frequent ED visits [2], hospitalizations and death [3,4].

Despite some disparity in the available data, nearly a third of patients are readmitted to hospital within 28 days of discharge [5], and one in five patients requires rehospitalization within 30 days of discharge after and admission for AECOPD [6,7]. Beyond this period of time, there is also a considerable variability in outcomes, with 90-day readmission rates ranging between 16% and 48% [8,9]. With regard to mortality after discharge, there appears to be a considerable variability related to samples [10]. On the other hand, it is worth mentioning that most databases of COPD episodes do no specifically specify if readmission is in hospitals or in EDs.

Factors that contribute to early readmission include premature discharge, poor discharge medication reconciliation, lack of family education on disease management and lack of communication with outpatient physicians [11,12,13,14,15]. Comorbidities are among factors that have also been identified as increasing risk for early readmissions after AECOPD [4]. The majority of people with COPD also have other medical problems, most commonly ischaemic heart disease. Moreover, at least more than half of patients admitted with AECOPD have one comorbidity, and a third of them will have four or more comorbidities [8,9]. It has been suggested that nearly a third of COPD admissions could be avoided through the implementation of evidence-based care interventions. A large body of evidence supports both pharmacological and nonpharmacological interventions to reduce risk of AECOPD and improves overall health status [5,16]. Nevertheless, once patients have been discharged from hospitals or EDs following an AECOPD, a specific plan to face and even to avoid future relapses is not routinely delivered [17]. Actually, despite the existence of clinical practice guidelines for COPD [1,18,19,20,21,22], there is a lack of consensus about discharge bundle interventions that should be implemented in order to reduce the risk of relapses [23].

## 2. Care Bundles

One example of an evidence-based intervention is the use of care bundles. The Institute of Healthcare Improvement defined care bundles as “a structured way of improving the processes of care and patients outcome”, based on a set of defined actions contributing to the achievement of a clearly specified aim [24,25]. Properly implemented protocol-based care bundles should enable teams to concentrate on a range of measurable activities and optimize certain associated outcomes [25].

Care bundles have an important role in quality improvement as they focus on best consistent and standardized practice, especially in common chronic conditions, such as asthma and heart failure [26,27]. Actually, they have been used in a variety of conditions with effectiveness demonstrated in a range of settings [16,28]. Ideally, the decision to include individual interventions in discharge bundles should be guided by the best evidence, clinical expertise and patient values [29]. Care bundles after discharging patients with COPD comprise a short list of evidence-based practices to be implemented prior to discharge for all patients admitted with this condition.

The effectiveness of AECOPD discharge care bundles is a relatively new area of research. In 2017, the British Thoracic Society has launched a COPD admission and discharge care bundles project with the aim of improving care and reducing readmissions for patients admitted for AECOPD. The project aims to provide independent evidence of the impact of COPD discharge care bundles on care during and after hospital admissions and future readmissions.

Data suggest that the AECOPD rehabilitation program and self-management can help to decrease exacerbations and improve the quality of live (QoL) of such patients [30,31]. Nevertheless, only individual evaluations of AECOPD discharge care bundles have been published in the scientific literature, and no formal synthesis of this body of evidence has been undertaken. In addition, to date, only a few randomized, controlled trials that specifically evaluate the effect of a comprehensive program with multidisciplinary input on patients on discharged after AECOPD are available [31,32,33,34].

## 3. Evidence

A recent systematic review on the effectiveness of AECOPD discharge care bundles found moderate evidence that their implementation is likely to reduce readmissions after AEs of COPD [2]. The review included 14 studies (five clinical trials, seven uncontrolled trials, and two interrupted time series). Primary outcomes were hospital readmissions and ED returns; secondary outcomes were mortality, physician visits, satisfaction of QoL of patients and economic outcomes.

It is worth remarking that some of those studies also included patients admitted by COPD and pneumonia [35,36] and patients admitted by COPD and heart failure [36,37].

The most common individual interventions included in discharge care bundles were: techniques of ensuring patients to access proper inhalers (nine studies); educational programs on self-management (nine studies); individually tailored care plans for self-management (eight studies); assessments/referrals for pulmonary rehabilitation (eight studies); arrangement of outpatient follow-ups (eight studies); and referrals to a smoking cessation program (seven studies) [2].

Interestingly, most of studies implemented the discharge care bundles in respiratory wards, with some internal medicine wards, and one study was conducted in EDs [34], which reported conflicting results in annual hospitalization trends before and after the implementation of the discharge bundles. Nevertheless, it was verified that the bundle acted as a focus for improving knowledge and delivering COPD care.

Meta-analysis of these randomized clinical trial data showed that discharge care bundles after hospital admissions for AECOPD led to fewer readmissions with no significant improvements in mortality and QoL [2].

## 4. Acute Exacerbations of Chronic Obstructive Pulmonary Disease Discharge Care Bundles

One of the first available studies published by Casas et al. in 2006 [38] demonstrated that an integrated care intervention prevents hospitalizations for AECOPD. In this study, patients were randomly treated with an individually tailored care plan upon discharge shared with the primary care team, as well as accessibility to a specialized nurse versus a usual care. Integrated care was associated with a lower rehospitalization during the follow-up of 12 months. The integrated plan included a comprehensive assessment of the patient at discharge, an educational program of self-management of the disease administered at discharge, an individually tailored care plan shared with a specialized nurse and primary care team and an accessibility of the specialized nurse to patient/carers and primary care professionals during the follow-up.

Hopkinson et al. [16] aimed to develop a pilot implementation of a COPD discharge care bundle in the setting of hospitalized patients. It comprised a checklist of five to six evidence-based practices that should be delivered to all patients. These care bundles were based on a previous published article [39], which proposed a checklist in eight targeted clinical areas, such as AECOPD. The study was performed in the setting of respiratory ward and the care bundle pack included:notification of the respiratory clinical nurse specialist of all admissions;supply of smoking cessation assistance if the patient is a smoker;assessment referrals for pulmonary rehabilitation;preparation of written information about COPD, including British Lung Foundation self-management booklets, oxygen alert cards and information about patient support groups.demonstration of patients’ satisfaction and use of inhalers;follow-up of appointments to be made with a specialist prior to discharge.

The 30-day readmission rate with the bundle used was 10.8% for patients, while the readmission rate without the bundle involved was 16.4%, demonstrating a moderate downward trend in readmission.

Laverty et al. [40] evaluated the impact of the AECOPD discharge care bundle on readmission following hospitalization. The care bundle included:smoking cessation;referral to pulmonary rehabilitation;appropriate education;assessment of patients understanding medication and use of inhalers.

The main finding of this study is that the implementation of the AECOPD discharge care bundle during hospital admission was associated with a change from an upwards trend in readmission rates for patients with COPD to a moderate downwards one; moreover, this study focused on readmissions to hospital, rather than other measures, such as mortality.

In another trial [33], patients were randomly selected to join in either the control (standard care) or the bundle group. In these groups, patients received next bundles, including:smoking cessation counseling;screening for gastroesophageal reflux disease;screening for a depression or anxiety;standardized inhaler education;a 48 h post-discharge telephone call.

The primary end point was the difference in the composite risk of hospitalizations or ED visits for AECOPD between these two groups in the 30 days after discharge. A secondary end point was the 90-day readmission rate. Pre-discharge bundle intervention in AECOPD was insufficient to reduce the 30-day risk of hospitalization or ED visits. Overall, the time of readmission in 30 and 90 days was similar between groups. In this study, the setting of COPD discharge bundle implementations was not reported.

A randomized trial by Ko et al. [32] focusing on the impact of AECOPD discharge bundles found a decrease in hospital readmissions and length of hospital stay at one-year follow-up, compared with subjects treated with the usual care and readmissions. Moreover, the QoL of the subjects at 12 months was improved. Bundles used at discharge included:smoking cessation program;screening for gastro-esophageal reflux;self-management of psychological distress and relaxation techniques;standardized inhaler education;a 3-month follow-up telephone call during one year.

## 5. Content of Key Findings of Acute Exacerbations of Chronic Obstructive Pulmonary Diseases Discharge Care Bundles

Nowadays, there is a lack of consensus on the specific measures that should be contained in an optimal discharge care bundle for patients with AECOPD.

Some authors [2] pointed out that differences in the content of bundles in the studies require an urgent need for comparative effectiveness research to guide future implementation of specific discharge care bundles, in order to optimize a community management in patients with AECOPD. A pioneering study recently proposed a patient-centered evidence-based and consensus-based discharge care bundle for patients with AEs of COPD. Presumably, this care bundle might ensure patients with COPD to be managed in proper and effective ways that were acceptable for them. This study described the deployment of a discharge care bundle for patients with COPD, based on evidence consensus among clinical experts and patients’ feedbacks using a modified Delphi method. In this study, the discharge bundle incorporated seven care terms in care bundles, which was an incorporation number proposed in previous recommendations [4], although other previous publications have proposed different numbers of individual interventions, with a median number of five care items per bundle [2]. Based on clinician and patient input, the seven unique care items in the COPD discharge bundle included in the consensus were:to ensure an adequate inhaler technique is demonstrated;to send discharge summary to family physicians and arrange follow-ups;to optimize and reconcile prescription of respiratory medications;to provide a written discharge management plan and assess patients and caregivers;to refer to pulmonary rehabilitation;to screen for frailty and comorbid conditions;to assess smoking status and refer to smoking cessation programme.

## 6. Could Admission Care Bundles Improve Results and Help Discharge from Emergency Departments?

As mentioned above, AECOPD are common presentation to EDs, and about 20% of AECOPD patients are discharged from EDs [41]. To our knowledge, only one study focused on AECOPD discharge care bundles in EDs [34]. Taking into account the increasing complexity of management and pressure on clinicians’ time in delivering acute medical care, the difficulty in delivering care bundles at discharge in EDs could be increased.

Interestingly, Graham et al. [34] studied the impact of a combined admission and COPD discharge care bundles on hospital mortality and readmission rates. Care bundles consisted of a seven-point acute care section on admission and a 10-point discharge section in 50 consecutive admissions. Despite the lack of full compliance, they found an improvement in mortality and readmissions, although the study did not specify the bundle care interventions.

Feasibility to implement discharge bundle care from EDs in AECOPD should be explored in the near future. In the meanwhile, a combination of admission and discharge care bundles in patients could be the key to start a new paradigm in the proper discharge of AECOPD patients from EDs.

## 7. Challenges of Implementing Discharge AECOPD Care Bundles

The design and implementation of care bundles after hospital discharge involves significant challenges [42]. Many barriers to implement interventions for patients with COPD have been described, and the main challenge seems to be related to the elements and interventions included in bundles and the strategies to their implementation [2,40]. When interventions are complex and incorporate different providers, successful translation is critical [43]. These difficulties would presumably have an impact on obtaining full compliance to bundle elements and thus would influence the final outcomes for those patients [42].

Moreover, other difficulties have been involved in the implementation of discharge AECOPD bundles. Among these, it is worth mentioning physicians’ ignorance about the potential benefit of a specific measure, such as pulmonary rehabilitation. This evidence-based practice that should be delivered to all patients, is frequently avoided, in particular among physicians who are not chest medicine specialists; the long waiting time reported for this service also plays a role in the lack of implementation of this measure [44]. The use of inhaler techniques, another evidence-based practice measure, seems to be difficult to achieve; actually, prescribers’ knowledge of inhalers and inhalation techniques remains poor, in particular among physicians not specialized in chest medicine [45].

In addition, despite the benefits of smoking cessation assistance, hospital physicians often think that this type of assistance is not included in their responsibilities [46]. Likewise, it is of note that despite guidelines recommendations [4] and advice given regarding influenza and pneumococcal vaccinations, their application as a part of bundle interventions remains limited.

Other described barriers are related to staffing. Too busy staff, staff shortages, lack of staff engagement, added workload by the bundle: all these factors seem to be critical for poor bundle implementation [42].

## 8. Conclusions

Preventing AECOPD, and thus breaking a cycle of recurrences seems to be crucial in COPD management. Evidence supports the need to improve transitions of care for patients with AECOPD across EDs, hospitals and community settings, and ensure coordination and continuity of care. Despite the lack of studies performed in the setting of EDs, data suggest that care bundles after discharging patients with AECOPD can help to decrease exacerbations, to reduce the risk of relapses and to improve the QoL of those patients. Efforts may be made to solve potential challenges of implementing discharge care bundles in patients with AECOPD. Further studies in the setting of ED are needed, in order to tailor specific discharge care bundles in those patients.

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
