# Peer review of "Care Bundles after Discharging Patients with Chronic Obstructive Pulmonary Disease Exacerbation from the Emergency Department"

_medsci, 2018, doi:10.3390/medsci6030063_

Round 1

Reviewer 1 Report

Lines 30 to 35, references to statistics are confusing due to different units used to explain readmissions and it does not specify if they are readmissions to emergency or to hospital. 

Even if authors considered published lists, on the final recommendations it is not indicated that screenings have to be taken into account. 

Author Response

Response to Reviewer comments. Thank You Very Much.

1-       Lines 30 to 35, references to statistics are confusing due to different units used to explain readmissions and it does not specify if they are readmissions to emergency or to hospital

I agree that the paragraph (line 30 tto 35) is confusing and units are mixed, therefore I appreciate your thorough review. The intention is basically to talk about readmissions. I agree that as regards mortality after admission due to COPD exacerbation, the available data are confusing; in consequence I will introduce few clarifications .

On the other hand,  We assume that could of interest to distinguish between hospital and emergency readmissions. Nevertheless, readmissions referred  in the articles reviewed discuss  in their majority  about the general term “hospital readmissions”. Unfortunately,  to date, satatistics database of COPD episodes mostly do not distinguish between readmission in emergency  or in hospital.

Reviewer 2 Report

I have read with interest the thorough review carried out by Dras. Gómez-Angelats and Sánchez, about the Care bundles after discharging patients with COPD exacerbation from the Emergency Department, in order to reduce readmissions, as well as mortality.

If we take into account that the exacerbation of COPD does not begin when the patient arrives at the Emergency Department, nor does it end after discharge, even after discharge from a hospitalization, we will understand that the management of the exacerbation should be maintained by part of the health professionals who care for the patient, from the beginning until the end, that is, until the patient returns to baseline. At this point, it is vital to connect all health professionals, such as the primary care physicians and nurses, doctors and nurses of the Emergency Department, as well as the pulmonologist and respiratory clinical nurse specialist.

After reading the present review, we see how there are similar points in all the studies reviewed. Among them, the coordination between the different assistance levels, the smoking cessation, the referral to Pulmonary Rehabilitation (as long as it is available), the verification and education in a correct inhalation technique, the education on the illness, as well as the structured action plans seem the most important, regardless of the severity of the exacerbation.

Despite the fact that most of the studies listed by the authors do not reach statistical significance, in all of them there seems to be a tendency to decrease readmissions, as well as an improvement in the quality of life, in the studies in which has analyzed. It is likely that studies with a greater number of subjects or with a more precise methodology and with greater external validity are needed.

It seems clear that whatever the severity of the exacerbation of copd, all patients should check the inhalation technique, smoking cessation, pulmonary rehabilitation (as long as it is available), appointment with Primary care, both doctor and nurse, and an early appointment with respiratory specialist.

A very important point is the social environment of the patient, so the assessment by social workers makes it an essential point at the time of discharge a patient of ED. It must be borne in mind that many patients live alone or without family support, which makes it very difficult for them to understand what the disease is or to assume the prescribed action plans. Therefore, in these cases, it is likely that residences of older people are a resource to be taken into account. We should do an educative work in these centers to the health personnel, in inhalation technique and education in the disease.

In conclusion, the review carried out by the authors of the manuscript is good, of high quality and with the messages issued clear and precise.

Author Response

We respond o the reviewer`s comments. Thank You Very much.

We greatly appreciate your comments. As you pointed out, most of the studies analyzed do no reach statistical significance and in many cases, the sampel is small and heterogeneous. We agree  that bundle care in  COPD after discharge shows  a tendency to decrease readmissions, an therefore It deserves further  research . All those results argue for improve methodological studies in this setting.